# Association between the saphenous vein diameter and venous reflux on computed tomography venography in patients with varicose veins

**Ji Yoon Choi**⬡*[⊛], **Ju-Hee Lee**[⊛], **Oh Jung Kwon**[⊛]

Division of Transplantation and Vascular surgery, Department of Surgery, Hanyang University Medical Center, Seoul, Korea

⊛ These authors contributed equally to this work.
* skytrio6@gmail.com

**Data Availability Statement:** All relevant data are within the manuscript and its Supporting Information files.

## Abstract

Three-dimensional computed tomography venography is a useful tool to identify increased saphenous vein diameter and provides a complementary road map for surgery in patients with varicose veins. In this study, we investigated the correlation between saphenous vein diameter on computed tomography venography and venous reflux detected on duplex ultrasonography. We enrolled 152 patients (213 extremities) who underwent endovenous laser ablation therapy, following high ligation of the saphenofemoral junction between January 2014 and December 2019. All patients underwent preoperative computed tomography venography evaluation. The saphenous vein diameter was measured on computed tomography venography, and venous reflux was evaluated in the operating room using Doppler ultrasonography. Among the 152 patients included in the study, 61 showed varicose veins affecting the bilateral extremities. Among the 213 extremities investigated, 165 (77.5%) and 48 (22.5%) extremities showed varicosities involving the greater and lesser saphenous veins, respectively. Among all extremities, venous reflux was detected in 172 (80.8%). The mean diameter of the greater saphenous vein measured 5 cm distal to the saphenofemoral junction was 8.07±1.82 mm in patients with reflux and 5.11±1.20 mm in patients without reflux (p < .05). The small saphenous vein diameter measured 5 cm distal to the saphenopopliteal junction was 7.65±1.74 mm in patients with reflux and 5.04±1.80 mm in patients without reflux (p < .05). Based on the receiver operating characteristic curve, the greater saphenous vein threshold diameter of 5.880 mm measured 5 cm distal to the saphenofemoral junction was the optimal cut-off value to predict reflux (sensitivity 91.4%, specificity 81.8%). The lesser saphenous vein diameter of 5.285 mm measured 5 cm distal to the saphenopopliteal junction was the optimal cut-off value to predict reflux (sensitivity 94.9%, specificity 75.0%). Vein diameter cannot be used as an absolute reference for venous reflux; however, it may have predictive value in patients with varicose veins. Computed tomography venography based measurements of vein diameter may serve as a useful diagnostic tool to predict venous reflux and recommend treatment.

**Funding:** The authors received no specific funding for this work.

**Competing interests:** The authors have declared that no competing interests exist.

## Introduction

Among all chronic venous diseases, varicose veins account for the most common type of lower extremity vein disorders with prevalence rates that vary between 5% and 30% [1,2]. Most patients visit outpatient clinics with lower extremity symptoms, including edema, pain, leg heaviness, and skin changes such as dermatitis, sclerosis, ulceration, and tortuous and dilated superficial veins, which occur secondary to volume overload in cutaneous veins due to valvular incompetence and blood flow abnormalities [3,4].

Treatment is aimed at eliminating venous reflux; therefore, duplex ultrasonography (DUS) is considered the gold standard to confirm the diameter of the dilated veins, venous reflux, the anatomical site of this abnormality, and the altered hemodynamics [1,5,6]. However, the results of DUS vary depending on the operator's skills, which serves as a limitation of this imaging modality.

Computed tomography venography (CTV) using three-dimensional (3D) reconstruction offers several advantages for diagnosis and optimal treatment planning [5,7,8]. CTV is useful to accurately delineate varicose vein anatomy, particularly in clinically challenging scenarios such as in patients with recurrent varicosities and provides a road map to guide the surgical procedure [7,9]. Therefore, our center uses CTV for preoperative evaluation of varicose veins.

Some studies have shown that reflux diagnosed on preoperative DUS reflects an increased saphenous vein diameter [1,2,10]. However, only a few reports have described the correlation between the saphenous vein diameter and US-proven reflux in patients with varicose veins.

We hypothesize that a statistically significant correlation exists between diameter of saphenous vein on CTV and pathological reflux on DUS. In this study, we investigated the correlation between saphenous vein diameter and reflux in patients with varicose veins to determine the cut-off diameter of the vein that can predict reflux based on preoperative CTV.

## Materials and methods

### Patients

The study was performed according to the Declaration of Helsinki, and Institutional Review Board approval was obtained. We conducted a retrospective, observational study using data extracted from medical records. The study protocol was approved by the institutional review board of our center, Republic of Korea (HYUH 2021-03-035), which waived the requirement for informed patient consent because of the retrospective nature of our study.

Between January 2014 and December 2019, 154 patients underwent varicose vein surgery of the lower extremities at our center. Of these, 152 patients were enrolled in this study, except for 2 patients with incomplete medical records. Demographic and clinical data such as age, sex, C category of the Clinical-Etiology-Anatomy-Pathophysiology classification, patients' symptoms, and site of involvement were retrospectively analyzed.

### Preoperative evaluation

All patients who visited the outpatient clinic with suspected varicose veins underwent careful evaluation with history-taking and physical examination, followed by preoperative 3D-CTV.

Preoperative CTV was obtained with a 64-channel helical CT scanner (Brilliance Scanner, Philips Healthcare) at a setting of 120 kVp and 230 mA. The scan parameters included 64×0.625 mm collimation and 3-mm slice thick reconstruction. CTV images were obtained after intravenous administration of a 150 mL nonionic iodinated contrast agent at a rate of 3.5 mL/s. Enhancement of deep and superficial veins, including varicose and perforating veins

was observed 3 min after contrast agent injection. All CTV images were reconstructed in the axial, coronal, and sagittal orientations.

## Operating room procedures

Intraoperatively, DUS was performed by a single surgeon using a duplex scanner (Samsung, HS60) with a 7.5 MHz linear probe after the patient was placed in the reverse Trendelenburg position(. Valvular function of the greater saphenous vein (GSV) was evaluated at the sapheno-femoral junction (SFJ) and that of the small saphenous vein (SSV) was evaluated at the level of the popliteal fossa. Reflux was diagnosed in patients with reflux time >0.5 s. Preoperative CTV was performed to measure the vein diameter 3 cm and 5 cm distal to the SFJ and saphenopo-pliteal (SPJ), and DUS was performed to confirm reflux at the same site to avoid false-positive vein reflux.

The CTV image was displayed on the screen intraoperatively to provide a roadmap for the surgeon. We performed high ligation of the saphenous vein and endovenous laser ablation (EVLA). Postoperatively, the patients' legs were wrapped using an elastic bandage, followed by elastic compression.

## Follow-up

Patients visited the outpatient clinic for 1-week follow-up for evaluation of the obliteration rate of the treated veins and complications.

## Statistical analysis

All statistical analyses were performed using the SPSS software, version 21.0 (SPSS, Chicago, IL, USA). Categorical variables were expressed as frequencies or percentages and were compared using the $X^2$ or the Fisher exact test. Continuous variables were expressed as means ±standard deviations and were compared using the Student's t test. After determination of the tendency, receiver operating characteristic (ROC) curve analysis was performed to confirm the optimal cut-off value of the saphenous vein diameter to predict reflux. A p value <0.05 was considered statistically significant.

## Results

We evaluated 213 extremities for venous reflux in 152 patients. Table 1 shows patient demographics. The GSV and SSV were evaluated in 165 (77.46%) and 48 (22.54%) extremities, respectively. The male:female ratio was 100:113 (46.9%:53.1%), and the mean patient age was 54.64±11.90 years (range 26–79 years). Venous reflux was detected in the GSV in 129 (78.2%) extremities and in the SSV in 43 (89.6%) extremities. CTV showed that the mean diameters of the GSV and SSV measured 5 cm distal to the SFJ and SPJ were 7.43±2.17 mm (3.0–14.8 mm) and 7.41±1.58 mm (3.05–11.68 mm), respectively.

In this study, patients were categorized into two groups based on the presence of venous reflux, and we performed an intergroup comparison of the saphenous vein diameter (Tables 2 and 3) based on the region of measurement. The GSV diameter measured 3 cm distal to the SFJ was 8.28±1.89 mm in patients with reflux and 5.634±1.41 mm in patients without reflux. The GSV diameter measured 5 cm distal to the SFJ was 8.07±1.82 mm in patients with reflux and 5.11±1.20 mm in those without reflux. The GSV diameter was significantly larger in both regions in patients with reflux (95% CI:-3.34,-1.95,t(163) = -7.49, p < .05 and 95% CI:-3.49,-2.44, t(163) = -11.96,p < .05). The SSV diameter measured 3 cm distal to the SPJ was 7.74 ±1.62 mm in patients with reflux and 4.85±0.57 mm in patients without reflux. The SSV

**Table 1. Demographics of all extremities which underwent operation of varicose veins.**

| | | All | | Great saphenous vein | | Small saphenous vein | |
|---|---|---|---|---|---|---|---|
| | | n = 213 | | n = 165 | | n = 48 | |
| Sex | | | | | | | |
| | Male | 100 | (46.9) | 76 | (46.1) | 24 | (50.0) |
| | Female | 113 | (53.1) | 89 | (53.9) | 24 | (50.0) |
| Age (years) | | 54.64±11.90 | | 55.18±11.83 | | 52.79±12.11 | |
| Height (cm) | | 163.86±8.72 | | 163.44±8.72 | | 165.32±8.64 | |
| Weight (kg) | | 67.32±11.68 | | 67.43±11.64 | | 66.93±11.93 | |
| Body mass index (kg/m$^2$) | | 24.79±4.19 | | 25.06±4.04 | | 23.85±4.52 | |
| Bilaterality | | 122 | (57.3) | 102 | (61.8) | 20 | (41.7) |
| Location | | | | | | | |
| | Right | 95 | (44.6) | 80 | (48.5) | 15 | (31.3) |
| | Left | 118 | (55.4) | 85 | (51.5) | 33 | (68.7) |
| HTN | | 47 | (22.1) | 35 | (21.2) | 12 | (25.0) |
| DM | | 21 | (9.9) | 15 | (9.1) | 6 | (12.5) |
| CAD | | 5 | (2.3) | 4 | (2.4) | 1 | (2.1) |
| CVA | | 13 | (6.1) | 11 | (6.7) | 2 | (4.2) |
| Cholesterol (mg/dL) | | 195.42±40.48 | | 198.26±39.53 | | 185.46±42.59 | |
| Triglyceride (mg/dL) | | 150.93±102.65 | | 151.04±98.23 | | 150.55±118.82 | |
| HDL-cholesterol (mg/dL) | | 57.70±51.81 | | 51.36±13.49 | | 77.97±10.06 | |
| LDL-cholesterol (mg/dL) | | 110.50±40.15 | | 109.96±37.67 | | 112.32±48.36 | |
| Onset of Symptom | | | | | | | |
| <1 year | | 84 | (39.9) | 61 | (37.0) | 23 | (47.9) |
| >1 year | | 129 | (60.1) | 104 | (63.0) | 25 | (52.1) |
| C classification | | | | | | | |
| | 1 | 4 | (1.9) | 3 | (1.8) | 1 | (2.1) |
| | 2 | 163 | (76.5) | 120 | (72.7) | 43 | (89.6) |
| | 3 | 30 | (14.1) | 28 | (17.0) | 2 | (4.2) |
| | 4 | 13 | (6.1) | 11 | (6.7) | 2 | (4.2) |
| | 5 | 1 | (0.5) | 1 | (0.6) | | |
| | 6 | 2 | (0.9) | 2 | (1.2) | | |
| Saphenous vein diameter below junction (mm) | | | | | | | |
| | 3cm | | | 7.74±2.10 | | 7.39±1.78 | |
| | 5cm | | | 7.43±2.17 | | 7.41±1.89 | |
| Presence of Reflux | | 172 | (80.8) | 129 | (78.2) | 43 | (89.6) |
| Complication | | 59 | (27.7) | 49 | (27.9) | 13 | (27.1) |

diameter measureed 5 cm distal to the SPJ was 7.65±1.74 mm in patients with reflux and 5.04 ±1.80 mm in those without reflux. The SSV diameter was significantly larger in both regions in patients with reflux (95% CI:-4.38,-1.41,t(46) = -3.93,p < .05 and 95% CI:-4.46,-0.76, t(46) = -2.85, p < .05). No significant intergroup differences were observed in the other characteristics.

ROC curves were used to determine the predictive value of venous reflux based on their location. With regard to the GSV (Fig 1), the optimal cut-off diameter that predicted reflux (92.2% sensitivity and 72.7% specificity) was 6.190 mm when the diameter was measured 3 cm distal to the SFJ and was 5.880 mm (91.4% sensitivity and 81.8% specificity) when the diameter was measured 5 cm distal to the SFJ. As shown in Fig 2, with regard to the SSV, the optimal

Table 2. Clinical characteristics of extremities according to the presence of reflux in great saphenous vein.

| | | All | Reflux negative | | Reflux positive | | p-value |
|---|---|---|---|---|---|---|---|
| | | n = 165 | n = 36 | | n = 129 | | |
| Sex | | | | | | | 0.176 |
| | Male | 76 | 13 | (36.1) | 63 | (48.8) | |
| | Female | 89 | 23 | (63.9) | 66 | (51.2) | |
| Age (years) | | 55.18±11.83 | 55.11±11.04 | | 55.19±12.01 | | 0.971 |
| Height (cm) | | 163.44±8.72 | 161.53±9.43 | | 163.96±8.48 | | 0.141 |
| Weight (kg) | | 67.43±11.64 | 65.11±14.87 | | 68.07±10.55 | | 0.178 |
| Body mass index (kg/m²) | | 24.79±4.19 | 24.14±5.72 | | 25.32±3.36 | | 0.20 |
| Bilaterality | | 102 | 25 | (69.4) | 77 | (59.7) | 0.287 |
| Location | | | | | | | 0.086 |
| | Right | 80 | 22 | (61.1) | 58 | (45.0) | |
| | Left | 85 | 14 | (38.9) | 71 | (55.0) | |
| HTN | | 35 | 12 | (33.3) | 23 | (17.8) | 0.044 |
| DM | | 15 | 1 | (2.8) | 14 | (10.9) | 0.136 |
| CAD | | 4 | 0 | 0 | 4 | (3.1) | 0.285 |
| CVA | | 11 | 1 | (2.8) | 10 | (7.8) | 0.290 |
| Cholesterol (mg/dL) | | 195.42±40.48 | 206.10±36.35 | | 196.07±40.23 | | 0.179 |
| Triglyceride (mg/dL) | | 150.93±102.65 | 135.96±81.62 | | 154.81±102.02 | | 0.423 |
| HDL-cholesterol (mg/dL) | | 57.70±51.81 | 54.70±12.42 | | 50.49±13.70 | | 0.216 |
| LDL-cholesterol (mg/dL) | | 110.50±40.15 | 98.55±52.40 | | 113.00±32.46 | | 0.252 |
| Onset of Symptom | | | | | | | 0.267 |
| | <1 year | 84 | 11 | (30.6) | 50 | (38.8) | |
| | >1 year | 129 | 25 | (69.4) | 79 | (61.2) | |
| C classification | | | | | | | 0.758 |
| | 1 | 3 | 1 | (2.8) | 2 | | |
| | 2 | 120 | 28 | (77.8) | 92 | | |
| | 3 | 28 | 4 | (11.1) | 24 | | |
| | 4 | 11 | 2 | (5.6) | 9 | | |
| | 5 | 1 | 0 | 0 | 1 | | |
| | 6 | 2 | 1 | (2.8) | 1 | | |
| Saphenous vein diameter below junction (mm) | | | | | | | |
| | 3cm | 7.74±2.10 | 5.63±1.41 | | 8.28±1.89 | | < .05 |
| | 5cm | 7.43±2.17 | 5.11±1.20 | | 8.07±1.82 | | < .05 |
| Complication | | 49 | 10 | (27.9) | 33 | (27.1) | 0.385 |

cut-off value that predicted reflux (sensitivity 94.9% and specificity 75.0%) was 5.285 mm when the diameter was measured 5 cm distal to the SPJ.

## Discussion

An increase in the prevalence of varicose veins has attracted much attention in the medical community, and research is being performed to gain a deeper understanding of the anatomy and hemodynamics of the venous system to ensure optimal treatment for this condition. DUS is considered the gold standard for preoperative evaluation of varicose veins, because it can provide both anatomical and functional assessment of the venous system [3,4,6,10,11]. SFJ or SPJ ligation and stripping constitutes standard treatment as reported by randomized trials that have shown good long-term results associated with this approach [12–14].

**Table 3. Clinical characteristics of extremities according to the presence of reflux in small saphenous vein.**

| | | All | Reflux negative | | Reflux positive | | p-value |
|---|---|---|---|---|---|---|---|
| | | n = 48 | n = 5 | | n = 43 | | |
| Sex | | | | | | | 0.637 |
| | Male | 24 | 2 | (40.0) | 22 | (51.2) | |
| | Female | 24 | 3 | (60.0) | 21 | (48.8) | |
| Age (years) | | 52.79±12.11 | 52.60±11.72 | | 52.81±12.29 | | 0.971 |
| Height (cm) | | 165.32±8.64 | 164.88±10.20 | | 165.36±8.62 | | 0.917 |
| Weight (kg) | | 66.93±11.93 | 61.48±7.88 | | 67.44±12.18 | | 0.344 |
| Body mass index (kg/m$^2$) | | 24.79±4.19 | 18.04±10.11 | | 24.52±3.12 | | 0.226 |
| Bilaterality | | 20 | 2 | (40.0) | 18 | (41.9) | 0.936 |
| Location | | | | | | | 0.143 |
| | Right | 15 | 3 | (60.0) | 12 | (27.9) | |
| | Left | 33 | 2 | (40.0) | 31 | (72.1) | |
| HTN | | 12 | 0 | | 12 | (27.9) | 0.178 |
| DM | | 6 | 0 | | 6 | (14.0) | 0.372 |
| CAD | | 1 | 0 | | 1 | (2.3) | 0.730 |
| CVA | | 2 | 0 | | 2 | *4.6) | 0.622 |
| Cholesterol (mg/dL) | | 185.46±42.59 | 211.98±66.36 | | 183.00±40.00 | | 0.196 |
| Triglyceride (mg/dL) | | 150.55±118.82 | 177.00±110.50 | | 147.71±121.23 | | 0.692 |
| HDL-cholesterol (mg/dL) | | 77.97±10.06 | 63.00±16.70 | | 79.63±107.56 | | 0.794 |
| LDL-cholesterol (mg/dL) | | 112.32±48.36 | 106.67±95.53 | | 113.00±43.01 | | 0.920 |
| Onset of Symptom | | | | | | | 0.200 |
| | <1 year | 23 | 4 | (80.0) | 19 | (44.2) | |
| | >1 year | 25 | 1 | (20.0) | 24 | (55.8) | |
| C classification | | | | | | | 0.885 |
| | 1 | 1 | 0 | | 1 | (2.3) | |
| | 2 | 43 | 5 | (100) | 38 | (88.4) | |
| | 3 | 2 | 0 | | 2 | (4.7) | |
| | 4 | 2 | 0 | | 2 | (4.7) | |
| | 5 | | | | | | |
| | 6 | | | | | | |
| Saphenous vein diameter below junction(mm) | | | | | | | |
| | 3cm | 7.39±1.78 | 4.85±0.57 | | 7.74±1.62 | | <.05 |
| | 5cm | 7.41±1.89 | 5.04±1.80 | | 7.65±1.74 | | <.05 |
| Complication | | 13 | 1 | (20.0) | 12 | (27.9) | 0.974 |

Interestingly an increasing number of hospitals are performing preoperative CTV routinely for evaluation in such cases, and a variety of endovenous treatment options, such as radiofrequency ablation (RFA), EVLA, and ultrasound-guided sclerotherapy are currently available.

Several studies have reported the usefulness of CTV in patients with varicose veins owing to its advantages [7,9,15]. A study performed by Kim [9] showed that CTV can serve as an excellent guide map for the treatment of varicose veins without significant complications and is useful for evaluation of perforators, anatomical variations, differential diagnosis of deep vein disease, and recurrence. Kim et al. [15] showed that CTV could provide information on SSV variations and reduce recurrence rates and intraoperative nerve injury. They focused on the location of saphenopopliteal junction(SPJ). SPJ morphology and the relationship between SSV and gastrocnemical vein and neural topography were important for correct removal of reflux

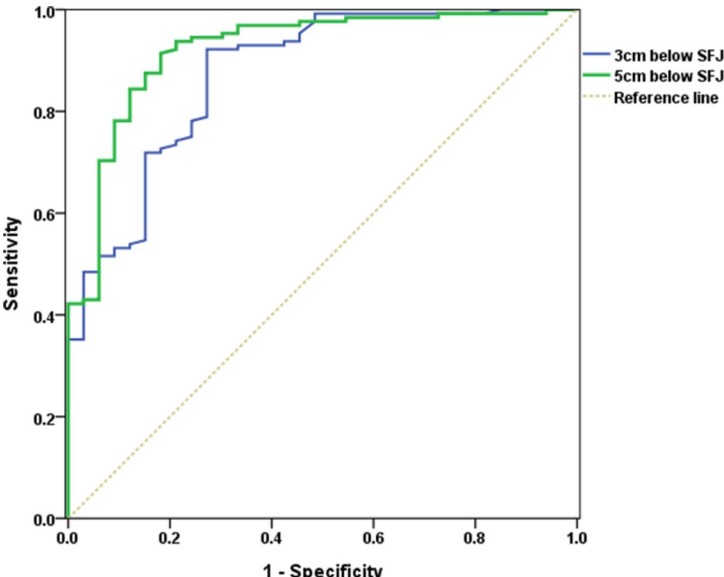

**Fig 1. Receiver operating characteristics curve to ascertain the GSV diameter for predicting the presence of reflux.**

mechanism and prevention of complication. They concluded that complete mapping of the venous networking, providing anatomical as well as hemodynamic data, was important for making decisions and surgical achievement.

Several recent studies have reported the safety and efficacy of EVLA with ligation of the saphenous vein as a safe and effective therapeutic strategy in patients with varicose veins. Imuzi et al. [12] reported the importance of high ligation of the saphenous vein as an essential step and showed that this approach was more effective than EVLA alone. This approach minimizes the risk of early recanalization of the treated saphenous veins, development of post-procedural deep venous thromboembolism, and recurrence.

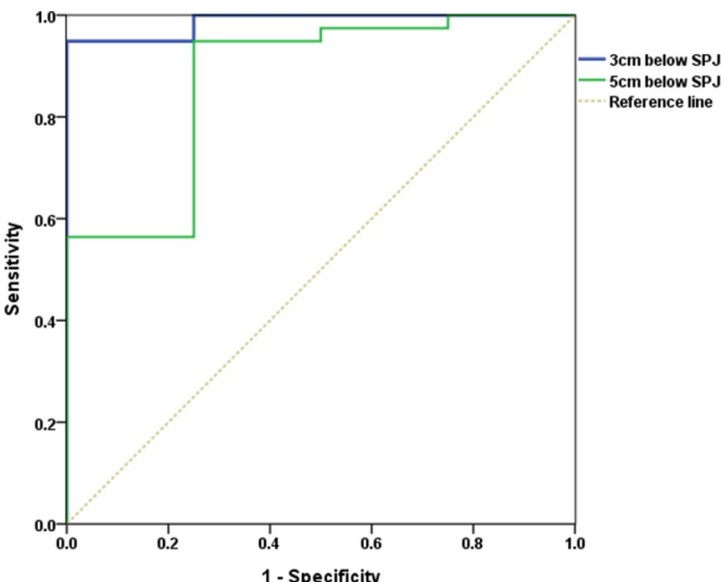

**Fig 2. Receiver operating characteristics curve to ascertain the SSV diameter for predicting the presence of reflux.**

Based on these reports, we performed CTV as a diagnostic tool for preoperative evaluation in patients with suspected varicose veins and EVLA combined with saphenous vein ligation as standard treatment.

Usually, the GSV measures 4 mm (or <3 mm) in diameter, and the SSV measures <3 mm in diameter. However, in patients with venous insufficiency, these veins are dilated (often significantly), and the diameter of the GSV with incompetent valves may even be >15 mm. Usually, significant reflux is obvious and is characterized by retrograde flow after releasing compression of a venous segment below the region being evaluated. Several studies have been performed to quantify the hemodynamic changes in varicose veins and to evaluate the morphological changes in the affected veins.

Lee et al. [16] observed that GSVs that showed insufficiency accompanied with varicosities were characterized by focal ectasia and diffuse dilatation >6 mm; CTV could predict GSV insufficiency with a sensitivity of 98.2% and specificity of 83.3%. Joh et al. [1] reported that a GSV threshold diameter of 5.05 mm (based on DUS) was the optimal cut-off value for prediction of reflux with 76% sensitivity and 60% specificity. With regard to the SSV, the cut-off diameter that predicted reflux was 3.55 mm with sensitivity and specificity of 87% and 71%, respectively. Navarro et al. [17] reported that GSV diameter >5.5 mm could predict abnormal reflux with sensitivity of 78% and specificity of 87%.

In our study, based on CTV evaluation, we observed that the GSV and SSV diameters were significantly greater in patients with reflux (p<0.05 and p<0.05), and 6.190 mm and 5.610 mm were the optimal cut-off diameters that predicted reflux in the GSV and SSV, respectively.

The saphenous vein diameter was measured at various regions of interest. The Union Internationale Phlebologie recommends that GSV diameter should be measured at two locations, 3 cm below the SFJ and at the proximal thigh (PT) [1,2,5]. However, in patients who undergo RFA or EVLA, the Cure Conservatrice et Hémodynamique de l'Insuffisance Veineuse en Ambulatoire CHIVA) group recommends that the GSV diameter be measured 15 cm distal to the SFJ because the PT site allows outcome assessment regardless of preservation of the GSV trunk [18]. Monzoda et al. [18] reported that measurements at the PT showed higher sensitivity and specificity to predict reflux and clinical class. Kim et al.[2] measured the GSV diameter at the SFJ, the mid thigh, lower thigh, and below-knee regions and observed that the GSV diameter measured at the lower thigh level was significantly greater than that measured at other sites and showed the highest area under the curve (AUC) value (0.642); the cutoff diameter for reflux was 5 mm ($p$ = .025).

In this study, we measured the diameter both 3 cm and 5 cm distal to the SFJ and the SPJ and observed that the GSV diameter measured 5 cm distal to the SFJ and SPJ showed a higher AUC value (0.922) with the cut-off diameter for reflux being 6.075 mm (p<0.05). The SSV diameter measured 3 cm below the aforementioned junctions showed a higher AUC value (0.987) with the cut-off diameter for reflux being 5.61 mm (p<0.05).

Following are the limitations of this study: (a) We measured the saphenous vein diameter using CTV because CTV is a relatively objective and non-operator-dependent diagnostic modality. We performed DUS after CTV to confirm reflux at the same level. However, this may have led to differences in diametrical positions in the same patient, which could have introduced an error in our results. Since CTV is performed while lying down on a bed and DUS is performed in reverse Trendelenburg position. Therefore, there is inevitably a difference in diameter depending on posture that causing of the vias. (b) This was a small-scale study, which was more severe when divided into GSV and SSV. Therefore, further large-scale studies are warranted. (c) Patients who underwent surgery for varicose veins did not undergo long-term postoperative follow-up; therefore, unavailability of long-term follow-up data is a

drawback of this research. Our results would be more convincing if we could perform long-term follow-up to determine complication or recurrence rates.

In addition, it is true that CT venography as an initial diagnostic tool has the advantage of objectively and clearly three-dimensional confirmation of anatomy, but also has clear disadvantages such as radiation exposure and nephrotoxicity due to the use of contrast materials. Therefore, studies to establish clear indications and guidelines to overcome the limitations will also be needed.

## Conclusion

In conclusion, vein diameter cannot be used as an absolute reference for venous reflux; however, it may show predictive value in patients with varicose veins. GSV diameters >6.190 mm and 5.880 mm when measured 3 cm and 5 cm, respectively distal to the SFJ were the optimal cutoff values for prediction of venous reflux. SSV diameters >5.610 mm and 5.285 mm when measured 3 cm and 5 cm, respectively distal to the SPJ were the optimal cutoff values for prediction of venous reflux. Therefore, if performed according to accurate indications and guidelines, CTV may serve as a valuable diagnostic tool for evaluation of vein diameter to predict reflux and recommend treatment and may also be useful during follow-up to monitor for recurrence in patients who undergo treatment.

## Supporting information

**S1 Data.**
(XLSX)

## Author Contributions

**Conceptualization:** Ji Yoon Choi, Oh Jung Kwon.

**Data curation:** Ji Yoon Choi, Ju-Hee Lee, Oh Jung Kwon.

**Formal analysis:** Ji Yoon Choi.

**Investigation:** Ju-Hee Lee.

**Methodology:** Ji Yoon Choi, Ju-Hee Lee.

**Supervision:** Ji Yoon Choi.

**Writing – original draft:** Ji Yoon Choi.

**Writing – review & editing:** Ji Yoon Choi.

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
