## [Decision Letter · Decision Letter 0]

4 Jun 2021

PONE-D-21-16623

Association between the saphenous vein diameter and venous reflux on computed tomography venography in patients with varicose veins

PLOS ONE

Dear Dr. Choi,

Thank you for submitting your manuscript to PLOS ONE. After careful consideration, we feel that it has merit but does not fully meet PLOS ONE’s publication criteria as it currently stands. Therefore, we invite you to submit a revised version of the manuscript that addresses the points raised during the review process.

Please address the issues and revise accordingly.

We look forward to receiving your revised manuscript.

Kind regards,

Academic Editor

PLOS ONE

Journal Requirements:

Reviewers' comments:

Reviewer's Responses to Questions

**Comments to the Author**

1. Is the manuscript technically sound, and do the data support the conclusions?

Reviewer #1: Partly

Reviewer #2: Yes

Reviewer #3: Partly

2. Has the statistical analysis been performed appropriately and rigorously? 

Reviewer #1: Yes

Reviewer #2: Yes

Reviewer #3: Yes

3. Have the authors made all data underlying the findings in their manuscript fully available?

Reviewer #1: Yes

Reviewer #2: Yes

Reviewer #3: Yes

4. Is the manuscript presented in an intelligible fashion and written in standard English?

Reviewer #1: Yes

Reviewer #2: Yes

Reviewer #3: Yes

5. Review Comments to the Author

Reviewer #1: 1. Abstract, page 2, line 27: How does CTV identify reflux? I understand measuring diameter, but it does not measure reflux.

2. Introduction, page 4, line 73: CTV as a roadmap for therapy seems like overkill to me. The saphenous vein is a straight vein and any anatomic variants are easily seen with ultrasound.

3. Introduction, page 4, line 77: As above, I do not understand what the authors mean by CTV-proven reflux. CTV images are static and do not demonstrate reflux.

4. Methods, page 5, line 110: Was the first duplex ultrasound on these patients performed intraoperatively? Did patients undergo preoperative duplex ultrasound to evaluate for reflux?

5. Methods, page 5, line 120: Were any adjunctive procedures, e.g., phlebectomy, sclerotherapy, performed, or just ablation.

6. Results, page 7, line 140: Was ablation performed in all patients? What was done when the intraoperative duplex did not show reflux?

7. Results, page 7, lines 150 and 155: What were the p values for the diameter differences? They are not listed in the text and in the tables it just states <0.05.

8. Discussion, page 9, line 185: How does preoperative CTV help to prevent intraoperative nerve injury?

9. Discussion, page 9, line 190: The authors cite one paper advocating high ligation in conjunction with ablation, but there is an abundance of literature in which ablation is done alone without ligation.

10. Discussion, page 9, line 196: What do the authors mean when they state “the GSV measures 4mm (or<3mm) in diameter”? This is not clear.

11. Discussion, page 9, line 208: Since the CT findings in this study are similar to reported duplex diameters in the literature for predicting reflux, what is the advantage of CT? It seems like an extra and unnecessary test to be done routinely.

12. Discussion, page 11, line 238: With respect to limitations, how consistent are venous diameters in the leg in serial CT exams? Veins can dilate and contract, so may be sensitive to volume status, temperature, etc.

Reviewer #2: 1. In the title nor the abstract, there is no indication of the study design. It has to be deduced from the text itself. It is not imminently discernible from the title or abstract if that is a retrospective or prospective design. That becomes clear in the material and methods section – line 86. I would suggest including the study's design in the title or abstract (as per STROBE). It might be good to include a prespecified hypothesis (e.g., we hypothesize that a positive statistically significant correlation exists between GSV/SSV diameter and pathological reflux on DUS) in the introduction section since the objective is presented clearly (»to determine the cut-off diameter of the vein that can predict reflux based on preoperative CTV«).

2. There is no clear indication of outcomes, exposures, predictors, potential confounders, and effect modifiers in the methods section. Those become clear in the results section. I would suggest defining them earlier – in the Methods section (as per STROBE).

3. How was the sample size arrived at? It can also be simply stated that this is a cohort under observation, and no power calculation was done.

4. In the results section (lines 144 – 156), there is a clear statement that GSV and SSV diameter was significantly larger in both regions in patients with reflux. I would suggest reporting the actual test statistics, e.g., a statistically significant difference in saphenous vein diameter of ... mm was found between both groups (95% CI, .. to ...), t(df) = ..., p < .05. From that writeup, it can be immediately deduced that a two-tailed test was used and that it was, e.g., an independent samples t-test.

5. A normal vein has some amount of reflux; it is physiological reflux. I would indicate (at least once) in the text that the reflux described here is pathological reflux. What is considered pathological reflux in this study is adequately stated (line 115). I would suggest including also an exact model of the ultrasound machine, not only an indication of the probe type.

Reviewer #3: This paper is a prospective clinical study that evaluated the contribution of CTV for evaluation of saphenous vein diameter as an adjunct for diagnosis and treatment of venous disease. The underlying premise is that duplex ultrasound, which is the gold standard for diagnosis of venous reflux, is operator dependent and may not provide the best road map in patients with aberrant anatomy/needing recurrent procedures. However, the benefit of CTV as used in this study is not clear to me and I have multiple concerns.

The patients in the study were first seen in clinic where an H&P was performed and then the patients were taken for a CTV. On the CTV, the GSV and SSV diameters were measured in 2 separate locations. It appears that a duplex ultrasound, which is the only way that venous reflux is evaluated in this study (cannot be evaluated on a single phase CT), was not performed until the patients were in the operating room. Venous duplex studies to evaluated reflux are standardly performed outpatient with no sedation as they are non-invasive procedures. It is unclear in the paper what the consent process would have been to take patients to the operating room without a study demonstrating whether or not the patient needed a procedure. Once the patients were in the operating room, only 77% of extremities were evaluated for GSV reflux and only 22% of extremities were evaluated for SSV reflux. It is not clear why 100% of extremities were not evaluated for both GSV and SSV reflux, as is standard. The treatment, which was high ligation and EVLA, was reasonable. It does not appear that the CTV findings changed the surgical management or workup of the patients; all patients underwent venous duplex and patients with reflux all obtained the same ablative procedure.

Overall, I do not know what the CTV to evaluate SSV and GSV diameter contributes to the care of venous disease in the study patients or would add to our treatment paradigm in venous patients. Although larger veins are more likely associated with reflux, smaller veins can also have symptomatic reflux. Additionally, for the diagnosis of venous reflux, a duplex is still necessary. Finally, venous duplex has the advantage of being a dynamic study that does not expose the patients to radiation or contrast.

Other specific concerns:

Introduction

1. For the sentence:

“Some studies have shown that reflux diagnosed on preoperative DUS reflects an increased saphenous vein diameter”

Reflux can be found in saphenous veins of any size – not sure what the authors are trying to say with this sentence. Please reword or remove.

Methods

1. For the sentence:

“We did not perform noncontrast CT in this study owing to concerns of medical radiation.”

You don’t need to explain why a non-contrast phase was not performed for a routine CTV.

2. Did all patients have normal preoperative renal function? Was this checked prior to the CTV?

3. Why do all of the DUS in the OR – doesn’t this mean an unnecessary OR trip for the patients who did not have reflux? Did patients get the DUS reflux study prior to the OR?

4. What anesthetic was used for the operative procedures?

5. Did CTV change the operative approach?

6. In postoperative follow up, how was vein obliteration and EHIT evaluated?

Results

1. How many patients had prior venous procedures? If so, what approach was used? Did this change what you saw on CT in terms of vein size?

2. Why was reflux only measured in 77% of extremities for GSVs and 22% of extremities for SSV? Conventionally, we evaluate for both in all patients when we are evaluating venous disease.

3. Were the CTV diameters compared to the duplex US diameters?

6. PLOS authors have the option to publish the peer review history of their article (what does this mean?). If published, this will include your full peer review and any attached files.

Reviewer #1: No

Reviewer #2: No

Reviewer #3: No

---

## [Author Response · Author response to Decision Letter 0]

5 Jul 2021

Letters to Editors & Reviewers

Dear Editor & Reviewers.

First of all, thank-you very much for your interest in my article.

As you have recommended me, I revised my article.

The details are followings,

Reviewer #1: 

1. Abstract, page 2, line 27: How does CTV identify reflux? I understand measuring diameter, but it does not measure reflux.

I have corrected what you mentioned.

This study investigates the relationship between the diameter measured by CT and the reflux confirmed in the US, and it has been modified accordingly.

2. Introduction, page 4, line 73: CTV as a roadmap for therapy seems like overkill to me. The saphenous vein is a straight vein and any anatomic variants are easily seen with ultrasound.

As you mentioned, it is also known that the US can identify anatomy. However, we focus on studies that obtain information about overall anatomy and variations, past surgical performance, and so on through CT venography.

3. Introduction, page 4, line 77: As above, I do not understand what the authors mean by CTV-proven reflux. CTV images are static and do not demonstrate reflux.

This study is a study to investigate the relationship between the diameter measured on CT and the reflux confirmed on the US, and the mentioned parts have been corrected.

4. Methods, page 5, line 110: Was the first duplex ultrasound on these patients performed intraoperatively? Did patients undergo preoperative duplex ultrasound to evaluate for reflux?

We measure reflux in some patients during outpatient visits and perform it in all patients in the operating room. Therefore, ultrasound findings measured in the operating room were used as evidence.

5. Methods, page 5, line 120: Were any adjunctive procedures, e.g., phlebectomy, sclerotherapy, performed, or just ablation.

We did not perform sclerotherpy as a treatment option. In a small number of cases, phlebectomy was performed on the thin branch, but the core and main treatment method was ablation.

6. Results, page 7, line 140: Was ablation performed in all patients? What was done when the intraoperative duplex did not show reflux?

When we decided to do surgery, it was based on the dilatation of the saphenous vein through CTV, but we also performed the surgery based on the patient's symptoms and preferences.Our center performed EVLA as the basic surgical method, so ablation was performed on all patients.

There were some patients who did not show reflux because the operation was performed if the patient complained of discomfort and strongly wanted to operate even if the dilation was not severe.

7. Results, page 7, lines 150 and 155: What were the p values for the diameter differences? They are not listed in the text and in the tables it just states <0.05.

Each p-value was accurately corrected and recorded in the results and table.

8. Discussion, page 9, line 185: How does preoperative CTV help to prevent intraoperative nerve injury?

They focused on the location of saphenopopliteal junction(SPJ). SPJ morphology and the relationship between SSV and gastrocnemical vein and neural topography were important for correct removal of reflux mechanism and prevention of complication. They concluded that complete mapping of the venous networking, providing anatomical as well as hemodynamical data, was important for making decisions and surgical achievement..

I added the above to the mentioned section.

9. Discussion, page 9, line 190: The authors cite one paper advocating high ligation in conjunction with ablation, but there is an abundance of literature in which ablation is done alone without ligation.

As you commented, EVLA without ligation is preferred. However, we could not ignore the potential risks such as early recanalization of treated saphenous vein, development of DVT after procedure , particularly in which patients with severe varicose veins. To minimize these risks, we performed saphenofemoral junction ligation.

10. Discussion, page 9, line 196: What do the authors mean when they state “the GSV measures 4mm (or<3mm) in diameter”? This is not clear.

We based on our reference group.

11. Discussion, page 9, line 208: Since the CT findings in this study are similar to reported duplex diameters in the literature for predicting reflux, what is the advantage of CT? It seems like an extra and unnecessary test to be done routinely.

This paragraph is intended to to mention how other papers have analyzed cutoffs showing reflux based on CT or doppler before explaining our CT based cutoff diameter.

12. Discussion, page 11, line 238: With respect to limitations, how consistent are venous diameters in the leg in serial CT exams? Veins can dilate and contract, so may be sensitive to volume status, temperature, etc.

What you mentioned may also be a variable, but overall, we thought that CTV was a test with less variation depending on the examiner than duplex.

Reviewer #2: 

1. In the title nor the abstract, there is no indication of the study design. It has to be deduced from the text itself. It is not imminently discernible from the title or abstract if that is a retrospective or prospective design. That becomes clear in the material and methods section – line 86. I would suggest including the study's design in the title or abstract (as per STROBE). It might be good to include a prespecified hypothesis (e.g., we hypothesize that a positive statistically significant correlation exists between GSV/SSV diameter and pathological reflux on DUS) in the introduction section since the objective is presented clearly (»to determine the cut-off diameter of the vein that can predict reflux based on preoperative CTV«).

Thanks for the good suggestion. I followed your recommendation and added it to the introduction.

2. There is no clear indication of outcomes, exposures, predictors, potential confounders, and effect modifiers in the methods section. Those become clear in the results section. I would suggest defining them earlier – in the Methods section (as per STROBE).

Thank you for your comments. I referred to it.

3. How was the sample size arrived at? It can also be simply stated that this is a cohort under observation, and no power calculation was done.

As described, this study is a retroactive study based on the medical records of patients who have been operated on in this hospital. Based on the sample sizes of several references, we first identified the number of EVLA cases implemented in recent years and analyzed them.

4. In the results section (lines 144 – 156), there is a clear statement that GSV and SSV diameter was significantly larger in both regions in patients with reflux. I would suggest reporting the actual test statistics, e.g., a statistically significant difference in saphenous vein diameter of ... mm was found between both groups (95% CI, .. to ...), t(df) = ..., p < .05. From that writeup, it can be immediately deduced that a two-tailed test was used and that it was, e.g., an independent samples t-test.

I think it is appropriate to describe the comparison value and the p-value.

5. A normal vein has some amount of reflux; it is physiological reflux. I would indicate (at least once) in the text that the reflux described here is pathological reflux. What is considered pathological reflux in this study is adequately stated (line 115). I would suggest including also an exact model of the ultrasound machine, not only an indication of the probe type.

I added the exact model of the ultrasound machine

Reviewer #3: 

This paper is a prospective clinical study that evaluated the contribution of CTV for evaluation of saphenous vein diameter as an adjunct for diagnosis and treatment of venous disease. The underlying premise is that duplex ultrasound, which is the gold standard for diagnosis of venous reflux, is operator dependent and may not provide the best road map in patients with aberrant anatomy/needing recurrent procedures. However, the benefit of CTV as used in this study is not clear to me and I have multiple concerns.

The patients in the study were first seen in clinic where an H&P was performed and then the patients were taken for a CTV. On the CTV, the GSV and SSV diameters were measured in 2 separate locations. It appears that a duplex ultrasound, which is the only way that venous reflux is evaluated in this study (cannot be evaluated on a single phase CT), was not performed until the patients were in the operating room. Venous duplex studies to evaluated reflux are standardly performed outpatient with no sedation as they are non-invasive procedures. It is unclear in the paper what the consent process would have been to take patients to the operating room without a study demonstrating whether or not the patient needed a procedure. Once the patients were in the operating room, only 77% of extremities were evaluated for GSV reflux and only 22% of extremities were evaluated for SSV reflux. It is not clear why 100% of extremities were not evaluated for both GSV and SSV reflux, as is standard. The treatment, which was high ligation and EVLA, was reasonable. It does not appear that the CTV findings changed the surgical management or workup of the patients; all patients underwent venous duplex and patients with reflux all obtained the same ablative procedure.

Thank-you for your comment.

First, this study was analyzed based on the medical records of patients who underwent EVLA with high ligation with varicose veins at our hospital from 2014 to 2019. Therefore, there are limitations in data verification and analysis.

In addition, preoperative ultrasound results were not included because only a limited number of patients had records of outpatient ultrasound, and all DUS was performed in the operating room, so the findings were based on intraoperative DUS findings. This is also mentioned in method.

In addition to the CTV findings and reflux, surgery was performed according to the patient's symptoms and the patient's wishes, so patients without reflux may have been included. Rather than analyzing whether it has a major impact on the CTV treatment paradigm, we wanted to check how much the diameter in the performed CT has a relationship with reflux and whether it is meaningful as a tool to predict reflux.

Other specific concerns:

Introduction

1. For the sentence:

“Some studies have shown that reflux diagnosed on preoperative DUS reflects an increased saphenous vein diameter”

Reflux can be found in saphenous veins of any size – not sure what the authors are trying to say with this sentence. Please reword or remove.

This paragraph is intended to mention that although many studies have been mentioned in terms of ultrasound, there are relatively few studies related to CTV.

Methods

1. For the sentence:

“We did not perform noncontrast CT in this study owing to concerns of medical radiation.”

You don’t need to explain why a non-contrast phase was not performed for a routine CTV.

Thanks for the recommendation. I deleted it because I thought it would be better to delete it as you mentioned.

2. Did all patients have normal preoperative renal function? Was this checked prior to the CTV?

We performed the basic lab before CTV in all patients, and performed CTV after confirming that the renal function was normal.

3. Why do all of the DUS in the OR – doesn’t this mean an unnecessary OR trip for the patients who did not have reflux? Did patients get the DUS reflux study prior to the OR?

In addition, preoperative ultrasound results were not included because only a limited number of patients had records of outpatient ultrasound, and all DUS was performed in the operating room, so the findings were based on intraoperative DUS findings. 

4. What anesthetic was used for the operative procedures?

We performed general anesthesia.

5. Did CTV change the operative approach?

We used CTV as a road map, and it did not affect the surgical method itself.

6. In postoperative follow up, how was vein obliteration and EHIT evaluated?

It was optionally performed according to the patient's symptoms at the outpatient visit after surgery, but it was not performed as a routine procedure.

Results

1. How many patients had prior venous procedures? If so, what approach was used? Did this change what you saw on CT in terms of vein size?

Patients with a history of ipsilateral surgery were excluded from this study.

There were 23 cases of patients who received treatment for varicose veins on the other side.

2. Why was reflux only measured in 77% of extremities for GSVs and 22% of extremities for SSV? Conventionally, we evaluate for both in all patients when we are evaluating venous disease.

We performed doppler US in the operating room for all the parts to be operated on. What you mentioned is the ratio of GSV to SSV among all enrolled patients.

3. Were the CTV diameters compared to the duplex US diameters?

Some patients had records, we lacked a record of measuring diameter with DUS.

I’ll waiting your final acceptance mail.

Thank-you

Kind Regards.

*Corresponding Author:

Ji Yoon Choi , MD,PhD.

Division of Transplantation and Vascular surgery, Department of Surgery, Hanyang University Medical Center, Seoul, Korea

E-mail : skytrio6@gmail.com

Telephone: 82-10-3374-5262

---

## [Decision Letter · Decision Letter 1]

14 Jul 2021

PONE-D-21-16623R1

Association between the saphenous vein diameter and venous reflux on computed tomography venography in patients with varicose veins

PLOS ONE

Dear Dr. Choi,

Thank you for submitting your manuscript to PLOS ONE. After careful consideration, we feel that it has merit but does not fully meet PLOS ONE’s publication criteria as it currently stands. Therefore, we invite you to submit a revised version of the manuscript that addresses the points raised during the review process.

Please address the issues and revise accordingly.

We look forward to receiving your revised manuscript.

Kind regards,

Academic Editor

PLOS ONE

Reviewers' comments:

Reviewer's Responses to Questions

**Comments to the Author**

1. If the authors have adequately addressed your comments raised in a previous round of review and you feel that this manuscript is now acceptable for publication, you may indicate that here to bypass the “Comments to the Author” section, enter your conflict of interest statement in the “Confidential to Editor” section, and submit your "Accept" recommendation.

Reviewer #1: (No Response)

Reviewer #2: (No Response)

2. Is the manuscript technically sound, and do the data support the conclusions?

Reviewer #1: No

Reviewer #2: Yes

3. Has the statistical analysis been performed appropriately and rigorously? 

Reviewer #1: Yes

Reviewer #2: Yes

4. Have the authors made all data underlying the findings in their manuscript fully available?

Reviewer #1: Yes

Reviewer #2: Yes

5. Is the manuscript presented in an intelligible fashion and written in standard English?

Reviewer #1: Yes

Reviewer #2: Yes

6. Review Comments to the Author

Reviewer #1: (No Response)

Reviewer #2: The authors addressed my concerns from 1 - 3 and 5 adequately.

Regarding comment No. 4, I cannot concur with the authors that the comparison value and p-value suffice. It is not good practice to state just the abovementioned values; p-value alone cannot tell readers the strength or size of an effect, change, or relationship. It should be avoided to report just the p values. It is recommended to provide the test statistics (t, F, U, etc.), correlation or regression coefficient (if applicable), or measure the effect size (https://www.editage.com/insights/the-correct-way-to-report-p-values). The above recommendations can also be found in most statistical textbooks - but with much more extended and elaborate explanations.

I strongly suggest reporting the actual test statistics, e.g., a statistically significant difference in saphenous vein diameter of ... mm was found between both groups (95% CI, .. to ...), t(df) =..., p < .05. From that writeup, it can be immediately deduced that a two-tailed test was used and that it was, e.g., an independent samples t-test.

7. PLOS authors have the option to publish the peer review history of their article (what does this mean?). If published, this will include your full peer review and any attached files.

Reviewer #1: No

Reviewer #2: No

---

## [Author Response · Author response to Decision Letter 1]

15 Jul 2021

Letters to Editors & Reviewers

Dear Editor & Reviewers.

First of all, I really appreciate for your interest and advice in my article.

As you have recommended me, I revised my article.

The details are followings,

Reviewer #2: The authors addressed my concerns from 1 - 3 and 5 adequately.

Regarding comment No. 4, I cannot concur with the authors that the comparison value and p-value suffice. It is not good practice to state just the abovementioned values; p-value alone cannot tell readers the strength or size of an effect, change, or relationship. It should be avoided to report just the p values. It is recommended to provide the test statistics (t, F, U, etc.), correlation or regression coefficient (if applicable), or measure the effect size (https://www.editage.com/insights/the-correct-way-to-report-p-values). The above recommendations can also be found in most statistical textbooks - but with much more extended and elaborate explanations.

I strongly suggest reporting the actual test statistics, e.g., a statistically significant difference in saphenous vein diameter of ... mm was found between both groups (95% CI, .. to ...), t(df) =..., p < .05. From that writeup, it can be immediately deduced that a two-tailed test was used and that it was, e.g., an independent samples t-test.

Thank-you for your comment.

As you suggested, I reviewed several statistical references.

The contents you mentioned were added to the result section and corrected, and the p-value discription of the abstract and table was also changed.

I’ll waiting your final acceptance mail.

Thank-you

Kind Regards.

*Corresponding Author:

Ji Yoon Choi , MD,PhD.

Division of Transplantation and Vascular surgery, Department of Surgery, Hanyang University Medical Center, Seoul, Korea

E-mail : skytrio6@gmail.com

Telephone: 82-10-3374-5262

---

## [Decision Letter · Decision Letter 2]

21 Jul 2021

PONE-D-21-16623R2

Association between the saphenous vein diameter and venous reflux on computed tomography venography in patients with varicose veins

PLOS ONE

Dear Dr. Choi,

Thank you for submitting your manuscript to PLOS ONE. After careful consideration, we feel that it has merit but does not fully meet PLOS ONE’s publication criteria as it currently stands. Therefore, we invite you to submit a revised version of the manuscript that addresses the points raised during the review process.

Please address the issues and revise accordingly.

We look forward to receiving your revised manuscript.

Kind regards,

Academic Editor

PLOS ONE

Reviewers' comments:

Reviewer's Responses to Questions

**Comments to the Author**

1. If the authors have adequately addressed your comments raised in a previous round of review and you feel that this manuscript is now acceptable for publication, you may indicate that here to bypass the “Comments to the Author” section, enter your conflict of interest statement in the “Confidential to Editor” section, and submit your "Accept" recommendation.

Reviewer #1: (No Response)

Reviewer #2: (No Response)

2. Is the manuscript technically sound, and do the data support the conclusions?

Reviewer #1: No

Reviewer #2: Yes

3. Has the statistical analysis been performed appropriately and rigorously? 

Reviewer #1: Yes

Reviewer #2: Yes

4. Have the authors made all data underlying the findings in their manuscript fully available?

Reviewer #1: Yes

Reviewer #2: Yes

5. Is the manuscript presented in an intelligible fashion and written in standard English?

Reviewer #1: Yes

Reviewer #2: Yes

6. Review Comments to the Author

Reviewer #1: 1. Since 20% of the patients in this series had negative duplex exams, how can the authors be sure based on CT alone that the procedure they underwent was necessary. Perhaps their symptoms were due to causes other than venous insufficiency. The authors should acknowledge this.

Reviewer #2: I congratulate the authors on their excellent work with the research. With the last revision, the results are reported accordingly to standard. I would suggest that they correct the write-up of the reported values of the test statistic (in their case, independent samples t-test or Student's t-test). The df, usually written in parentheses after t, represents the degrees of freedom; df = N-2, number of all cases in the group - 2).

So in their case, it should be written for GSV; t(163) ...

And for SSV, it should be t(46) ... That would be the usual way to write it.

E.g., from the text:

The GSV diameter was significantly larger in both regions in patients with reflux (95% CI, -3.34 to -1.95), t(163)=-7.49 , p< .05 and (95% CI, -3.49 to -2.44), t(163)=-11.96, p< .05.

7. PLOS authors have the option to publish the peer review history of their article (what does this mean?). If published, this will include your full peer review and any attached files.

Reviewer #1: No

Reviewer #2: No

---

## [Author Response · Author response to Decision Letter 2]

21 Jul 2021

Dear Editor & Reviewers.

First of all, thank-you very much for your interest in my article.

As you have recommended me, I revised my article.

The details are followings,

Comments to the Author

2. Is the manuscript technically sound, and do the data support the conclusions?

Reviewer #1: No

Thank you. First, as described above, this study is a retrospective study based on the medical records of patients who underwent surgery at this hospital. The number of EVLA cases implemented in recent years was first identified and analyzed based on the sample size of several references. This part is described in the discussion as a limitation of this study.

6. Review Comments to the Author

Reviewer #1: 1. Since 20% of the patients in this series had negative duplex exams, how can the authors be sure based on CT alone that the procedure they underwent was necessary. Perhaps their symptoms were due to causes other than venous insufficiency. The authors should acknowledge this.

I agree with your comment. However, we initially tried to rule out whether their symptoms or dilatation of saphenous veins are due to other structural causes (DVT or vascular abnormality, etc) through CTV when patients first visited the outpatient clinic. If there was no structural abnormality mentioned above as a result of the CTV, it was considered to be due to venous insufficiency.

Reviewer #2: I congratulate the authors on their excellent work with the research. With the last revision, the results are reported accordingly to standard. I would suggest that they correct the write-up of the reported values of the test statistic (in their case, independent samples t-test or Student's t-test). The df, usually written in parentheses after t, represents the degrees of freedom; df = N-2, number of all cases in the group - 2).So in their case, it should be written for GSV; t(163) ...

And for SSV, it should be t(46) ... That would be the usual way to write it.

E.g., from the text:

The GSV diameter was significantly larger in both regions in patients with reflux (95% CI, -3.34 to -1.95), t(163)=-7.49 , p< .05 and (95% CI, -3.49 to -2.44), t(163)=-11.96, p< .05.

Thank-you for your comment.

As you suggested, I corrected the contents..

I’ll waiting your final acceptance mail.

Thank-you

---

## [Decision Letter · Decision Letter 3]

28 Jul 2021

PONE-D-21-16623R3

Association between the saphenous vein diameter and venous reflux on computed tomography venography in patients with varicose veins

PLOS ONE

Dear Dr. Choi,

Thank you for submitting your manuscript to PLOS ONE. After careful consideration, we feel that it has merit but does not fully meet PLOS ONE’s publication criteria as it currently stands. Therefore, we invite you to submit a revised version of the manuscript that addresses the points raised during the review process.

Please revise accordingly.

We look forward to receiving your revised manuscript.

Kind regards,

Academic Editor

PLOS ONE

Journal Requirements:

Reviewers' comments:

Reviewer's Responses to Questions

**Comments to the Author**

1. If the authors have adequately addressed your comments raised in a previous round of review and you feel that this manuscript is now acceptable for publication, you may indicate that here to bypass the “Comments to the Author” section, enter your conflict of interest statement in the “Confidential to Editor” section, and submit your "Accept" recommendation.

Reviewer #1: (No Response)

Reviewer #2: All comments have been addressed

Reviewer #4: All comments have been addressed

2. Is the manuscript technically sound, and do the data support the conclusions?

Reviewer #1: No

Reviewer #2: (No Response)

Reviewer #4: Yes

3. Has the statistical analysis been performed appropriately and rigorously? 

Reviewer #1: Yes

Reviewer #2: (No Response)

Reviewer #4: Yes

4. Have the authors made all data underlying the findings in their manuscript fully available?

Reviewer #1: Yes

Reviewer #2: (No Response)

Reviewer #4: Yes

5. Is the manuscript presented in an intelligible fashion and written in standard English?

Reviewer #1: Yes

Reviewer #2: (No Response)

Reviewer #4: Yes

6. Review Comments to the Author

Reviewer #1: (No Response)

Reviewer #2: (No Response)

Reviewer #4: All comment have been addressed. The article is well written. With the last revision the suggestions have been corrected.

7. PLOS authors have the option to publish the peer review history of their article (what does this mean?). If published, this will include your full peer review and any attached files.

Reviewer #1: No

Reviewer #2: No

Reviewer #4: No

---

## [Author Response · Author response to Decision Letter 3]

15 Aug 2021

Dear Editor & Reviewers.

First of all, I really appreciate for your interest and advice in my article.

Journal Requirements:

I reviewed all references and there were no retracted articles.

I really thank-you for your concern.

I’ll waiting your final acceptance mail.

Thank-you

Kind Regards.

---

## [Decision Letter · Decision Letter 4]

27 Sep 2021

PONE-D-21-16623R4Association between the saphenous vein diameter and venous reflux on computed tomography venography in patients with varicose veinsPLOS ONE

Dear Dr. Choi,

Thank you for submitting your manuscript to PLOS ONE. After careful consideration, we feel that it has merit but does not fully meet PLOS ONE’s publication criteria as it currently stands. Therefore, we invite you to submit a revised version of the manuscript that addresses the points raised during the review process.

Please address the issues and revise accordingly.

We look forward to receiving your revised manuscript.

Kind regards,

Academic Editor

PLOS ONE

Reviewers' comments:

Reviewer's Responses to Questions

**Comments to the Author**

1. If the authors have adequately addressed your comments raised in a previous round of review and you feel that this manuscript is now acceptable for publication, you may indicate that here to bypass the “Comments to the Author” section, enter your conflict of interest statement in the “Confidential to Editor” section, and submit your "Accept" recommendation.

Reviewer #2: (No Response)

Reviewer #5: (No Response)

2. Is the manuscript technically sound, and do the data support the conclusions?

Reviewer #2: Yes

Reviewer #5: Yes

3. Has the statistical analysis been performed appropriately and rigorously? 

Reviewer #2: Yes

Reviewer #5: Yes

4. Have the authors made all data underlying the findings in their manuscript fully available?

Reviewer #2: Yes

Reviewer #5: Yes

5. Is the manuscript presented in an intelligible fashion and written in standard English?

Reviewer #2: Yes

Reviewer #5: Yes

6. Review Comments to the Author

Reviewer #2: I congratulate the authors on their excellent work with the research. With the last revision, the results are reported almost accordingly to standard. I would suggest that they correct the write-up of the reported values of the test statistic (in their case, independent samples t-test or Student's t-test). The df, usually written in parentheses after t, represents the degrees of freedom; df = N-2, number of all cases in the group - 2). So in their case, it should be written for GSV; t(163) ...

And for SSV, it should be t(46) ... That would be the usual way to write it.

E.g., from the text:

The GSV diameter was significantly larger in both regions in patients with reflux (95% CI, -3.34 to -1.95), t(163)=-7.49, p< .05 and (95% CI, -3.49 to -2.44), t(163)=-11.96, p< .05.

I am sending the exact same comment as with the previous review. In the submitted text degrees of freedom are still reported as t(df) and not as the actual value. (Explanation is above - including an actual example from the text.) If it is not corrected it simply looks a bit odd. Please do correct it before publishing.

Reviewer #5: Thank you for your paper dealing with the association between SV diameter and venous reflux. Here are my comments:

1. I have a serious ethical problem with your study design and your practice: which is the reason to perform an additional CT scan in a patient with varicose veins and to expose him or her to radiation? Looking at the range of the age, you have performed a completely unnecessary CT scan in patients at the age of 29. Why?

2. You are mentioning that there is an increasing number of hospitals performing preop CTV. Firstly, I miss any references in this statement in your discussion. Secondly, which is the rationale behind this practice? And if you are taking so important informations from the CTV, which it will be extremely difficult to convince me, why not MRV?

3. Although it is not important for the outcome of the study, why are you performing high ligation and EVLA and not only EVLA closely to the junction?

4. Finally, the reflux of a vein is diagnosed in a standing position. During the CTV the patient is lying on a bed. How can you associate diameters and reflux having the patient in different positions?

7. PLOS authors have the option to publish the peer review history of their article (what does this mean?). If published, this will include your full peer review and any attached files.

Reviewer #2: No

Reviewer #5: **Yes: **Theodosios Bisdas

---

## [Author Response · Author response to Decision Letter 4]

30 Sep 2021

Dear Editor & Reviewers.

First of all, thank-you very much for your interest in my article.

As you have recommended me, I revised my article.

The details are followings,

Comments to the Author

6. Review Comments to the Author

Reviewer #2: I congratulate the authors on their excellent work with the research. With the last revision, the results are reported almost accordingly to standard. I would suggest that they correct the write-up of the reported values of the test statistic (in their case, independent samples t-test or Student's t-test). The df, usually written in parentheses after t, represents the degrees of freedom; df = N-2, number of all cases in the group - 2). So in their case, it should be written for GSV; t(163) ...

And for SSV, it should be t(46) ... That would be the usual way to write it.

E.g., from the text:

The GSV diameter was significantly larger in both regions in patients with reflux (95% CI, -3.34 to -1.95), t(163)=-7.49, p< .05 and (95% CI, -3.49 to -2.44), t(163)=-11.96, p< .05.

I am sending the exact same comment as with the previous review. In the submitted text degrees of freedom are still reported as t(df) and not as the actual value. (Explanation is above - including an actual example from the text.) If it is not corrected it simply looks a bit odd. Please do correct it before publishing.

Thank-you for your comment.

It was my mistake that I didn't correct it exactly as you said.

As you suggested, I corrected the contents..

Reviewer #5: Thank you for your paper dealing with the association between SV diameter and venous reflux. Here are my comments:

Thank-you for your overall comment.

Some of our related limitations and reasons are described below.

1. I have a serious ethical problem with your study design and your practice: which is the reason to perform an additional CT scan in a patient with varicose veins and to expose him or her to radiation? Looking at the range of the age, you have performed a completely unnecessary CT scan in patients at the age of 29. Why?

Thank-you for your comment

Your comments are also related to the limitations of this study. As described, this study is a retroactive study based on the medical records of patients who have been operated on in this hospital. 

And, we initially tried to rule out whether their symptoms or dilatation of saphenous veins are due to other structural causes (DVT or vascular abnormality, etc) through CTV when patients first visited the outpatient clinic. If there was no structural abnormality mentioned above as a result of the CTV, it was considered to be due to venous insufficiency. We also obtain information about overall anatomy and variations, past surgical performance, and so on through CTV. We performed the basic lab before CTV in all patients, and performed CTV after confirming that the renal function was normal.

2. You are mentioning that there is an increasing number of hospitals performing preop CTV. Firstly, I miss any references in this statement in your discussion. Secondly, which is the rationale behind this practice? And if you are taking so important informations from the CTV, which it will be extremely difficult to convince me, why not MRV?

While we were looking for a reference, we became interested in the papers that mention the utility of CTV (Reference No.7,9,15). And we mentioned this part at the beginning of the introduction and discussion. 

We performed the surgery according to the patient's symptoms and the patient's wishes, in addition to the CTV findings and reflux. As mentioned above, we performed CTV to rule out other diseases, then we used CTV as a road map, and it did not affect the surgical method itself. So, we wanted to check how much the diameter in the performed CT has a relationship with reflux and whether it is meaningful as a tool to predict reflux. rather than analyzing whether it has a major impact on the CTV treatment paradigm. 

And we did not consider carry out MRV because the biggest disadvantage of MRV is that it is expensive.

3. Although it is not important for the outcome of the study, why are you performing high ligation and EVLA and not only EVLA closely to the junction?

Thank-you for your concern.

I know that many center prefer EVLA without ligation is as you mentioned. However, we could not ignore the potential risks such as early recanalization of treated saphenous vein, development of DVT after procedure , particularly in which patients with severe varicose veins. We searched the literature for surgical methods that can minimize recurrence, and the above methods were considered. To minimize the recurrence rate, we started saphenofemoral junction ligation.

4. Finally, the reflux of a vein is diagnosed in a standing position. During the CTV the patient is lying on a bed. How can you associate diameters and reflux having the patient in different positions?

We measure reflux in some patients during outpatient visits and perform it in all patients in the operating room. Therefore, ultrasound findings measured in the operating room were used as evidence. Preoperative ultrasound results were not included because only a limited number of patients had records of outpatient ultrasound, and all DUS was performed in the operating room, so the findings were based on intraoperative DUS findings. This is also mentioned in method.

I’ll waiting your final acceptance mail.

Thank-you

Kind Regards.

---

## [Decision Letter · Decision Letter 5]

24 Nov 2021

PONE-D-21-16623R5Association between the saphenous vein diameter and venous reflux on computed tomography venography in patients with varicose veinsPLOS ONE

Dear Dr. Choi,

Thank you for submitting your manuscript to PLOS ONE. After careful consideration, we feel that it has merit but does not fully meet PLOS ONE’s publication criteria as it currently stands. Therefore, we invite you to submit a revised version of the manuscript that addresses the points raised during the review process. Please revise.

 Please submit your revised manuscript by Jan 08 2022 11:59PM. If you will need more time than this to complete your revisions, please reply to this message or contact the journal office at plosone@plos.org. Please include the following items when submitting your revised manuscript:A rebuttal letter that responds to each point raised by the academic editor and reviewer(s). You should upload this letter as a separate file labeled 'Response to Reviewers'.A marked-up copy of your manuscript that highlights changes made to the original version. You should upload this as a separate file labeled 'Revised Manuscript with Track Changes'.An unmarked version of your revised paper without tracked changes. You should upload this as a separate file labeled 'Manuscript'.

We look forward to receiving your revised manuscript.

Kind regards,

Academic Editor

PLOS ONE

Reviewers' comments:

Reviewer's Responses to Questions

**Comments to the Author**

1. If the authors have adequately addressed your comments raised in a previous round of review and you feel that this manuscript is now acceptable for publication, you may indicate that here to bypass the “Comments to the Author” section, enter your conflict of interest statement in the “Confidential to Editor” section, and submit your "Accept" recommendation.

Reviewer #2: All comments have been addressed

Reviewer #5: All comments have been addressed

2. Is the manuscript technically sound, and do the data support the conclusions?

Reviewer #2: Yes

Reviewer #5: No

3. Has the statistical analysis been performed appropriately and rigorously? 

Reviewer #2: Yes

Reviewer #5: Yes

4. Have the authors made all data underlying the findings in their manuscript fully available?

Reviewer #2: Yes

Reviewer #5: Yes

5. Is the manuscript presented in an intelligible fashion and written in standard English?

Reviewer #2: Yes

Reviewer #5: Yes

6. Review Comments to the Author

Reviewer #2: (No Response)

Reviewer #5: I still cannot accept your response about the need for preoperative CT in young patients with varicose veins. It is not acceptable to perform CT in young patients with venous insufficiency just to see the anatomy, previous operations etc. These are no indications to justify exposure to radiation. If a patient has a arteriovenous or venovenous malformation is of course indicated, but none of your patients had such a problem. Finally, your argumentation regarding my final comment about the measurement of the vein diameter in the CT, where the patient is lying on the bed is too poor.

7. PLOS authors have the option to publish the peer review history of their article (what does this mean?). If published, this will include your full peer review and any attached files.

Reviewer #2: No

Reviewer #5: **Yes: **Theodosios Bisdas

---

## [Author Response · Author response to Decision Letter 5]

29 Nov 2021

Reviewer #5: I still cannot accept your response about the need for preoperative CT in young patients with varicose veins. It is not acceptable to perform CT in young patients with venous insufficiency just to see the anatomy, previous operations etc. These are no indications to justify exposure to radiation. If a patient has a arteriovenous or venovenous malformation is of course indicated, but none of your patients had such a problem. Finally, your argumentation regarding my final comment about the measurement of the vein diameter in the CT, where the patient is lying on the bed is too poor.

There has been a change in the composition of the vascular surgeon during the period included in the study.

Previous vascular surgeons, since the 2010s, at our hospital, doppler US selectively, CT venography have been performed for several years for patients visiting outpatients with varicose veins. Recently, after becoming two vascular surgeons including myself, Doppler US is mainly performed first. This study referred to studies on the relationship between diameter and reflux after Doppler US, and studies related to CT venography and varicose veins. And, we started to compare the meaning of CT venography performed at our center. As you said, there were no cases of arteriovenous malformations in enrolled patients, but it was helpful to check the overall structure and other diseases before surgery, the shape of the recurrent varicose vein, etc., and to refer to it as a roadmap during surgery. However, even if the evaluation of renal function has been performed in consideration of the risk of contrast agents, I fully agree with the risks and limitations of radiation exposure that you mentioned. Therefore, we are in a state of control to some extent regarding the use of CTV as a diagnostic method.

I will explain the part mentioned about how to measure Diameter. As mentioned, it is common to measure reflux in a standing state. However, as previously mentioned, when referring to the medical records, the presence or absence of reflux was not evaluated on the outpatient basis in all patients, and all patients were evaluated for the presence or absence of reflux in the operating room with reverse Trendelenburg position. Therefore, we used reflux findings confirmed in the operating room as a comparison item. Since CTV is performed while lying down on a bed, I agree with your opinion that there is inevitably a difference in diameter depending on posture. At first, the analysis of the diameter of the Doppler US measured in the operating room was also considered, but there were also missing data. Therefore, the diameter of CTV in all patients was analyzed. It was measured by changing from the Supine state to the reverse Trendelenburg position, but I thought that the difference in this posture could be the cause of the vias, so I added this to the limitation. 

Overall, as you said, there are limitations to our data, but we tried to use it as much as possible to analyze and find meaning.

We look forward to your understanding and good news.

Thank-you

Kind Regards.

---

## [Decision Letter · Decision Letter 6]

6 Dec 2021

PONE-D-21-16623R6Association between the saphenous vein diameter and venous reflux on computed tomography venography in patients with varicose veinsPLOS ONE

Dear Dr. Choi,

Thank you for submitting your manuscript to PLOS ONE. After careful consideration, we feel that it has merit but does not fully meet PLOS ONE’s publication criteria as it currently stands. Therefore, we invite you to submit a revised version of the manuscript that addresses the points raised during the review process. Please revise.

We look forward to receiving your revised manuscript.

Kind regards,

Academic Editor

PLOS ONE

Journal Requirements:

Reviewers' comments:

Reviewer's Responses to Questions

**Comments to the Author**

1. If the authors have adequately addressed your comments raised in a previous round of review and you feel that this manuscript is now acceptable for publication, you may indicate that here to bypass the “Comments to the Author” section, enter your conflict of interest statement in the “Confidential to Editor” section, and submit your "Accept" recommendation.

Reviewer #2: All comments have been addressed

Reviewer #5: All comments have been addressed

2. Is the manuscript technically sound, and do the data support the conclusions?

Reviewer #2: Yes

Reviewer #5: Yes

3. Has the statistical analysis been performed appropriately and rigorously? 

Reviewer #2: Yes

Reviewer #5: Yes

4. Have the authors made all data underlying the findings in their manuscript fully available?

Reviewer #2: Yes

Reviewer #5: Yes

5. Is the manuscript presented in an intelligible fashion and written in standard English?

Reviewer #2: Yes

Reviewer #5: Yes

6. Review Comments to the Author

Reviewer #2: I read the comments of other reviewers. I do have to concur that doing CTV for chronic venous insufficiency without further possible indication is a highly dubious practice - it is also outside of all recommendations that I am aware of. The ethical committee that I am a member of would never grant agreement for such a study in a prospective manner. However, as you explained, you were doing retrospective (medical record) research. The CTV was not of your primary indication, and as far as I can understand, the Ethical committee agreed for such a retrospective review - I can imagine that prospective design would not be granted. With the acknowledgment that this is a retrospective design, my concerns about ethical issues are not acute (also with an acknowledgment that the Ethical committee decided on that beforehand). If this were a prospective design study - my recommendation would be to reject it due to ethical issues.

With the methodology of measuring the saphenous vein diameter - to me, it is interesting finding that there was such a big difference between refluxing and non-refluxing veins - even in the supine position (as measured per CTV). Since you did the Student t-test, it might be useful to report Cohen's d point estimate that would indicate the effect size. In my view, these findings are worth following up with US study, perhaps comparison between standing diameter and supine diameter difference compared to pathological reflux (present/absent). It might be that relative difference between standing and supine diameter will be bigger with refluxing as compared to non-refluxing veins (or vice-versa)?

Reviewer #5: Thank you for your response. Now it makes more sense and I hope that this is the real story behind. However, you have to admit all this information about the old and the new policy regarding CT venography in young patients with varicose veins and to make it crystal clear in your limitations, that this is not the way to go for preoperative scanning. Otherwise, I am sorry but you will never receive green light from me.

7. PLOS authors have the option to publish the peer review history of their article (what does this mean?). If published, this will include your full peer review and any attached files.

Reviewer #2: No

Reviewer #5: **Yes: **Theodosios Bisdas

---

## [Author Response · Author response to Decision Letter 6]

12 Jan 2022

6. Review Comments to the Author

Reviewer #2: I read the comments of other reviewers. I do have to concur that doing CTV for chronic venous insufficiency without further possible indication is a highly dubious practice - it is also outside of all recommendations that I am aware of. The ethical committee that I am a member of would never grant agreement for such a study in a prospective manner. However, as you explained, you were doing retrospective (medical record) research. The CTV was not of your primary indication, and as far as I can understand, the Ethical committee agreed for such a retrospective review - I can imagine that prospective design would not be granted. With the acknowledgment that this is a retrospective design, my concerns about ethical issues are not acute (also with an acknowledgment that the Ethical committee decided on that beforehand). If this were a prospective design study - my recommendation would be to reject it due to ethical issues.

With the methodology of measuring the saphenous vein diameter - to me, it is interesting finding that there was such a big difference between refluxing and non-refluxing veins - even in the supine position (as measured per CTV). Since you did the Student t-test, it might be useful to report Cohen's d point estimate that would indicate the effect size. In my view, these findings are worth following up with US study, perhaps comparison between standing diameter and supine diameter difference compared to pathological reflux (present/absent). It might be that relative difference between standing and supine diameter will be bigger with refluxing as compared to non-refluxing veins (or vice-versa)?

Thank-you for your recommendation.

When checking the raw data of this study, the number of US findings confirmed in outpatient clinic and operating rooms is small, and in particular, the number of diameters measured in the standing state is small. Therefore, I think it will be difficult to analyze the part you mentioned with the current data. However, it would be good to analyze the difference in diameter according to posture through prospective research through ultrasound .

Thank you for a good idea.

Reviewer #5: Thank you for your response. Now it makes more sense and I hope that this is the real story behind. However, you have to admit all this information about the old and the new policy regarding CT venography in young patients with varicose veins and to make it crystal clear in your limitations, that this is not the way to go for preoperative scanning. Otherwise, I am sorry but you will never receive green light from me.

Thanks for your understanding and further comment.

The limitations you mentioned are additionally mentioned at the end of the previous limitation.

In addition, it was added to the conclusion section that it should be performed according to accurate indications and guidelines.

---

## [Decision Letter · Decision Letter 7]

21 Jan 2022

Association between the saphenous vein diameter and venous reflux on computed tomography venography in patients with varicose veins

PONE-D-21-16623R7

Dear Dr. Choi,

We’re pleased to inform you that your manuscript has been judged scientifically suitable for publication and will be formally accepted for publication once it meets all outstanding technical requirements.

Kind regards,

Academic Editor

PLOS ONE

Additional Editor Comments (optional):

Reviewers' comments:

Reviewer's Responses to Questions

**Comments to the Author**

1. If the authors have adequately addressed your comments raised in a previous round of review and you feel that this manuscript is now acceptable for publication, you may indicate that here to bypass the “Comments to the Author” section, enter your conflict of interest statement in the “Confidential to Editor” section, and submit your "Accept" recommendation.

Reviewer #2: All comments have been addressed

Reviewer #5: All comments have been addressed

2. Is the manuscript technically sound, and do the data support the conclusions?

Reviewer #2: Yes

Reviewer #5: Yes

3. Has the statistical analysis been performed appropriately and rigorously? 

Reviewer #2: Yes

Reviewer #5: Yes

4. Have the authors made all data underlying the findings in their manuscript fully available?

Reviewer #2: Yes

Reviewer #5: Yes

5. Is the manuscript presented in an intelligible fashion and written in standard English?

Reviewer #2: Yes

Reviewer #5: Yes

6. Review Comments to the Author

Reviewer #2: (No Response)

Reviewer #5: (No Response)

7. PLOS authors have the option to publish the peer review history of their article (what does this mean?). If published, this will include your full peer review and any attached files.

Reviewer #2: No

Reviewer #5: **Yes: **Theodosios Bisdas

---

## [Editor Report · Acceptance letter]

3 Feb 2022

PONE-D-21-16623R7 

Association between the saphenous vein diameter and venous reflux on computed tomography venography in patients with varicose veins 

Dear Dr. Choi:

I'm pleased to inform you that your manuscript has been deemed suitable for publication in PLOS ONE. Congratulations! Your manuscript is now with our production department. 

Kind regards, 

on behalf of

Dr. Robert Jeenchen Chen 

Academic Editor

PLOS ONE